# Technologies for the Instrumental Evaluation of Physical Function in Persons Affected by Chronic Obstructive Pulmonary Disease: A Systematic Review

**DOI:** 10.3390/s22176620

**Published:** 2022-09-01

**Authors:** Alberto Zucchelli, Simone Pancera, Luca Nicola Cesare Bianchi, Alessandra Marengoni, Nicola Francesco Lopomo

**Affiliations:** 1Department of Information Engineering, Università degli Studi di Brescia, Brescia 25123, Italy; 2Aging Research Center, Department of Neurobiology, Care Sciences and Society, Karolinska Institutet, Solna SE-171 65, Sweden; 3IRCCS Fondazione Don Carlo Gnocchi, Milan 20148, Italy; 4Department of Clinical and Experimental Sciences, Università degli Studi di Brescia, Brescia 25123, Italy

**Keywords:** COPD, technology, instrumental evaluation, physical function

## Abstract

Several systems, sensors, and devices are now available for the instrumental evaluation of physical function in persons with Chronic Obstructive Pulmonary Disease (COPD). We aimed to systematically review the literature about such technologies. The literature search was conducted in all major scientific databases, including articles published between January 2001 and April 2022. Studies reporting measures derived from the instrumental assessment of physical function in individuals with COPD were included and were divided into application and validation studies. The quality of validation studies was assessed with the Consensus-based Standards for the selection of health Measurement Instruments (COSMIN) risk of bias tool. From 8752 articles retrieved, 21 application and 4 validation studies were included in the systematic review. Most application studies employed accelerometers, gait analysis systems, instrumented mattresses, or force plates to evaluate walking. Surface electro-myography or near-infrared spectroscopy were employed in four studies. Validation studies were heterogeneous and presented a risk of bias ranging from inadequate to doubtful. A variety of data regarding physical function can be retrieved from technologies used in COPD studies. However, a general lack of standardization and limitations in study design and sample size hinder the implementation of the instrumental evaluation of function in clinical practice.

## 1. Introduction

Chronic obstructive pulmonary disease (COPD) is a respiratory condition characterized by a high prevalence and a strong association with disability and mortality; it has been estimated that almost 10% of the worldwide population [1] is affected by this disease and that, in high-income countries, COPD is the fourth leading cause of death [2].

The evaluation of physical performance and function, by investigating—for example—the reduction in self-walked distance, is pivotal to the clinical management of persons affected by COPD, as suggested by current guidelines [3]. Several mechanisms may impact the physical performance of persons with COPD, for example, impaired ventilatory mechanics (such as dynamic hyperinflation), modification of the ventilation–perfusion relationship and hypoxemia, pulmonary hypertension, and other cardiovascular factors. The 6 min walking test (6MWT) is currently implemented to measure the impact of the COPD on physical performance. However, physical function tests may also help to gain important information about the risk of exacerbation and the impact of the disease on several organs and systems, returning a global measure of health [4,5]. A strong link between physical functioning and quality of life has also been reported [6].

The technological advancements achieved in the last decades led to the development of several systems, devices, and sensors that might be used to instrumentally evaluate physical function and performance. Exploiting such solutions has two main advantages. In first place, it increases the reliability of the measurements, by limiting the intra- and inter-operator variability [7,8]. Secondly, it can be used to obtain an objective measurement of physical function characteristics that are only seldom qualitatively evaluated in current practice, for example, measures of symmetry and balance can be acquired while performing a 6 min walking test [9]. Tri-axial accelerometers [9], high-speed cameras and markerless motion capture systems [10], and instrumented mattresses and force plates [11], as well as surface electromyography (sEMG) [12], are devices that may be suitable for the evaluation of physical function in persons affected by COPD. However, these devices considerably differ one from the other in terms of performance, price, dimension, and the tests implemented for the instrumental evaluation of physical performance.

Although previous reviews reported on the application of specific tools for the assessment of subjects with COPD, to date, no systematic reviews provided a comprehensive perspective on the application of the technological solutions for the functional evaluation of their physical performance in clinical and research settings [9,13,14].

Therefore, we aimed to systematically review the available scientific literature about the implementation of technologies—including systems, devices, and sensors—for the evaluation of physical function in subjects presenting COPD.

## 2. Methods

In accordance with PRISMA guidelines, the protocol for this systematic review was registered with the Open Science Framework (OSF) on 7 December 2020 (DOI:10.17605/OSF.IO/GXW6S; https://osf.io/gxw6s (accessed on 7 December 2020)) [15]. The article was written in keeping with the Guidelines for Meta-Analyses and Systematic Reviews of Observational Studies (MOOSE).

### 2.1. Eligibility Criteria and Information Sources

We reviewed studies providing information about the instrumental evaluation of physical function in persons with a diagnosis of COPD. We included studies that (1) addressed either a function (such as walking) or a test of physical performance (such as the 6-Meter Walking Test—6MWT), (2) studies that reported quantitative measures derived from the instrumental assessment of physical function, and (3) studies conducted either in a clinical or a laboratory setting.

Case-series studies, letters to the editor, reviews, and metanalyses, as well as commentaries, were excluded from this systematic review. Studies evaluating an isolated movement or characteristic (e.g., hand-grip strength), a specific body part, non-functional activities (e.g., cycling), or that reported only the assessment of respiratory function were also excluded. Finally, we excluded studies that performed remote/home-based measurements (e.g., studies assessing physical activity using an accelerometer worn at home for several days were excluded). This review was limited to articles in English or Italian, published between January 2001 and April 2022. The literature search was conducted in PubMed, Web of Science, Embase, and Scopus. An ethical committee’s approval was not needed for the conduction of this study.

### 2.2. Search Strategy

The search strategy for PubMed (available in the Appendix A) was adapted from the search filter proposed by a previous study [16]. In brief, the search string included terms for (1) construct search, (2) population search, and (3) instrument search. The search included terms related to physical function (e.g., walking, gait, balance, etc.), the population of interest (e.g., chronic obstructive pulmonary disease), and the instrument used for the assessment (e.g., actigraphy, kinematic analysis, and wearable sensors). Then, these searches were combined with the search filter for measurement properties and the exclusion filter to remove irrelevant records. The search terms were subsequently adapted for use with the other databases. References and additional files from selected articles were checked to identify further studies eligible for inclusion. The search was re-run just before the final analysis to retrieve new studies suitable for inclusion.

### 2.3. Selection Process

Two researchers (A.Z. and S.P.) independently screened the articles’ titles and abstracts after duplicate exclusion. Conflicts were resolved via consensus. In case a consensus was not reached, a third assessor (NFL) was included in the discussion. The full texts from all selected abstracts were retrieved, and they were screened and selected using the same procedure. An online application [17] was used to simplify the process of abstract and full-text screening. Study characteristics and information were independently extracted from selected papers by two assessors. Extracted data were then compared and possible inconsistencies were resolved.

### 2.4. Data Items and Presentation of Results

We collected data on (1) the study (i.e., author, year, country, and design), (2) participants (i.e., sample size, demographic, and clinical characteristics), and (3) instrumental evaluation (i.e., device features, application procedures, and provided parameters). Then, the selected studies were divided into two groups: the first one included those employing technology-derived metrics to describe the functional characteristics of COPD participants, to compare healthy and COPD participants, or to investigate the association between physical function and other health-related outcomes. This group of studies was named “application studies”. The second group, including articles aiming to evaluate the performance (e.g., reliability, measurement error, precision, and validity) of the instrumental evaluation of physical function, was named “validation studies”.

### 2.5. Risk of Bias Assessment

We assessed the risk of bias for the validation studies employing the Consensus-based Standards for the selection of health Measurement Instruments COSMIN Risk of Bias tool [18]. The tool offers two different sets of criteria for studies’ evaluation according to their aim (reliability studies and measurement error studies). The worst-score-count method was applied to determine the risk of bias. The risk of bias was independently evaluated by the two reviewers: conflicts were resolved by the third assessor.

## 3. Results

A total of 8752 articles were retrieved from the literature search. Out of these, 24 were included in the present study, as shown in Figure 1. A total of 21 articles were considered application studies [10,11,12,19,20,21,22,23,24,25,26,27,28,29,30,31,32,33,34,35,36], whereas 4 were defined as validation studies [23,37,38,39] (Table 1). One study was included in both groups due to its double aim [23].

**Figure 1 sensors-22-06620-f001:**
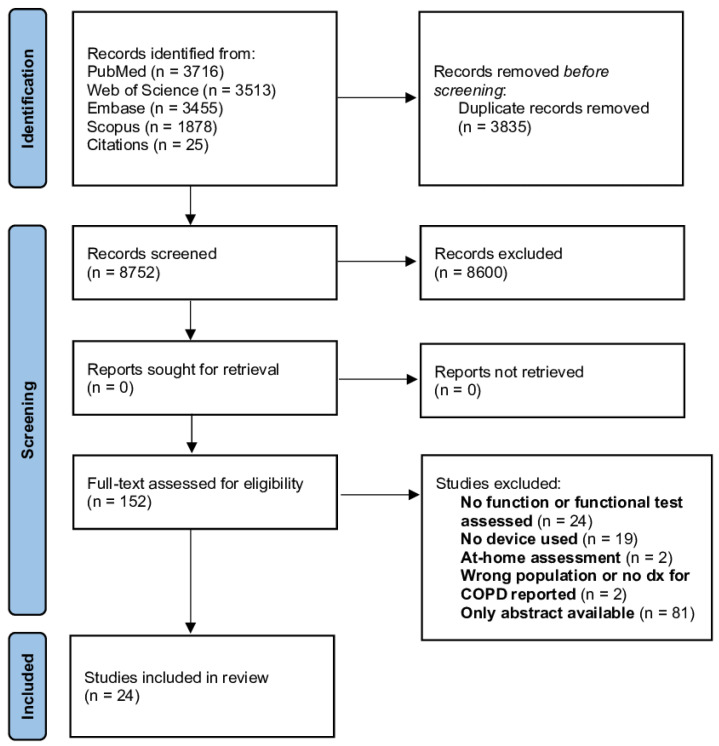
Flow diagram of the included studies.

COPD, chronic obstructive pulmonary disease.

### 3.1. Study Characteristics

Most of the studies (*n* = 22) presented an observational design, and 18 aimed at comparing the functional characteristics of participants with COPD with those of individuals without this condition (mostly healthy controls). The number of persons with COPD included in the selected studies ranged between 6 and 80, with a proportion of female patients comprised between 0% and 83% and a mean age comprised between 60 and 72 years old. The mean % of predicted Forced Expiratory Volume in the 1 s (FEV1) ranged from 35.1% to 58.4%. A recent COPD exacerbation was reported as an exclusion criterion in nine studies, whereas hypoxemia and/or chronic oxygen supplementation were considered exclusion criteria by four studies. The presence of comorbidities able to interfere with physical function tests was reported as an exclusion criterion in most studies (*n* = 18). Most studies (*n* = 17) reported that the diagnosis of COPD was made in accordance with the Global Initiative for Chronic Obstructive Lung Disease (GOLD) guidelines, whereas the others mainly reported that the diagnosis was made by a physician or reported in the medical charts.

### 3.2. Application Studies

Eleven application studies employed a single device for the evaluation of physical function, whereas in the remaining papers, multiple instruments were used, as shown in Table 2 and Table 3. Accelerometers (either alone or in combination with other devices) were employed in five studies, whereas force plates or instrumented mattresses were used in eight papers. The employment of surface electromyography (sEMG) was reported in five studies. Gait analysis systems, three-dimensional (3D) motion captures systems, or high-speed cameras were employed in seven studies, and two studies reported the utilization of near-infrared spectroscopy (NIRS). Walking was the most commonly assessed function (15 studies). The other application studies evaluated a variety of tests and functions: 6 min step test, pegboard and ring test, balance and perturbation tests, and domestic activities of daily living (simulated in the laboratory setting).

Table 4 shows the parameters derived from the employment of each technology: accelerometers were used mainly to obtain information about activity intensity (i.e., n. counts/unit of time) or volume (i.e., total n. counts or steps), although some spatiotemporal parameters were inferred from accelerometric data (step length, step time, gait speed). Spatiotemporal parameters of gait were mainly obtained using instrumented mattress and gait analysis systems. Force plates were employed in a variety of tests (among which, sit-to-stand tests, perturbed balance tests, and jump tests). The parameters obtained from such technologies were strongly related to the test performed: they ranged from the duration of meaningful function segments (such as the time taken for standing from the sitting position) to inferred measures of muscle power. Surface EMGs and NIRSs were employed to investigate specific muscles’ activation time, duration, signal intensity, or infer oxygen consumption during the execution of functional tests.

### 3.3. Validation Studies

Four studies were classified as validation studies (Table 1). As shown in Table 5, validation studies were heterogeneous in terms of technology employed and aim. Walking parameters were evaluated using 6MWT in two studies [37,38], using a 5 min walking protocol in one study [39] and a 10 m walkway in the remaining one [23]. Three [23,37,38] out of four studies measured gait speed, whereas the remaining one [39] assessed walking distance. One study [37] estimated gait speed using a mobile phone application based on an accelerometric measurement and compared such measure with manual walkway testing. Another study [38] investigated the validity of an instrumented treadmill with virtual reality and 3D motion analysis system evaluation by comparing the results with overground 6MWT. A third paper [23] evaluated the test–retest reliability of gait speed assessment exploiting a tri-axial accelerometer. Lastly, one study [39] investigated the correlation between the measurement obtained from a motion sensor and criterion methods during the assessment of walking distance at two different speeds.

### 3.4. Risk of Bias (Validation Studies)

According to the COSMIN tool, study quality was doubtful in three studies [23,38,39] and inadequate in one study [37]. Unclear concealed administration of tests and score assignment, or different conditions between repeated measurements, were the main reasons contributing to the doubtful quality of the studies. One study [37] was considered of inadequate quality due to important flaws in the sample composition.

## 4. Discussion

In this systematic review, we evaluated the implementation of different technologies—including systems, devices, and sensors—to assess the physical function of persons affected by COPD. We found that a wide variety of technologies, including small wearable devices (such as accelerometers), as well as cumbersome gait analysis systems, were employed in the selected studies. In most cases, these technological solutions were implemented for evaluation purposes, whereas a limited number of trials aimed to validate or assess the reliability of different tools in the specific context given by patients with COPD; unfortunately, this subset of studies showed doubtful or inadequate quality.

### 4.1. Same Metrics and Different Technologies

In our systematic review, we found that similar metrics were retrieved using a variety of technologies. For example, the spatiotemporal parameters of gait were obtained from accelerometers, gait analysis systems, and instrumented mattresses. In addition to the importance of maintaining the ecology of movement, containing operating costs and ensuring its applicability in clinical contexts, the validity of measurements obtained by different technologies is of pivotal importance for the implementation of an instrumental evaluation of mobility. Instrumented mattresses and walkway of force plates have been reported as valid technologies for the assessment of spatiotemporal parameters in healthy individuals when compared with video-based systems as gold standards [40,41]. The spatiotemporal parameters of gait obtained from a tri-axial accelerometer located near the center of gravity or from instrumented mattresses have also been shown to exhibit good-to-excellent collinearity in healthy individuals [42,43]. In addition, a recent meta-analysis [8] showed that the validity of Inertial Measurement Unit (IMU)-derived spatiotemporal parameters was generally excellent (using either instrumented walkway, instrumented mattresses, or motion capture systems as reference). However, the authors of the latter study [8] highlighted how this finding was strongly limited by the quality of the investigated studies, generally characterized by low statistical power. Our systematic review extends the issue of the absence of high-quality studies investigating the validity of the instrumental evaluation of physical function to the setting of COPD. For example, only one study [39] included in this systematic review compared measurements obtained from an accelerometer with a video-based assessment of gait: the collinearity between the two methodologies for the measurement of simple characteristics of gait (total distance walked and intensity) was moderate-to-low (Pearson’s rho ranging between 0.47 and 0.63). It is also worth mentioning that it is likely that the validity of an instrumented evaluation of function in persons affected by chronic conditions is lower than the one reported in healthy individuals. A recent study [44] comparing gait events recognition obtained from magneto-IMUs and instrumented mattresses, for example, showed that the errors in the estimation of the initial contact, stride time, and step time were significantly lower for healthy older adults in comparison with participants affected by Parkinson’s disease. This result is likely to be explained by the higher heterogeneity that can be found in pathological patterns of gait and the consequent difficulty in finding rules and algorithm for the identification of gait events. However, a previous review of this topic was not able to define the characteristic traits of gait patterns in individuals with COPD due to contrasting results in the existing literature [14].

However, the impact of possible measurement errors in the instrumental evaluation of physical function in COPD-related clinical or research practice is unknown. Almost the totality of the application studies included in our systematic review were cross-sectional, mostly with a case-control design; the possible association of specific technology-derived metrics with meaningful clinical outcome was not investigated, as well as possible confounders that may explain the differences found in terms of instrumental metrics between persons with COPD and healthy controls. It is crucial, for the implementation of technologies in clinical practice, to comprehensively investigate the possible associations between technology-derived metrics and both poor-health-related outcomes and specific indicators of disease progression in COPD.

### 4.2. The Instrumental Evaluation of the 6 min Walking Test

Out of eighteen studies (including both application and validation ones) evaluating gait, nine were conducted by performing the 6MWT. This result is not surprising given the current importance of this test protocol in COPD clinical practice [3]. The 6MWT, performed either on a treadmill or on the ground, was used to investigate a variety of parameters, exploiting different technologies; accelerometers, instrumented mattresses, gait analysis system, and sEMG were used to retrieve gait volume and intensity, spatiotemporal parameters, and muscle activation and fatigue.

Such novel technologies allow retrieving information about the variability of specific parameters or metrics while the test is performed: for example, a previous work has shown that fatigue may impact the temporal variability of spatiotemporal parameters [45] or that the distance walked may significantly impact gait rhythmicity [46].

Rutkowski and colleagues [32] reported that, both in participants with COPD and healthy controls, stride lengths were higher at the beginning of the test in comparison with those measured after 3 min. They also reported that stride length seemed to revert to baseline values at the end of the 6MWT, probably because of the provided instructions and motivation. Liu and colleagues [26] reported the coefficients of variation (i.e., standard deviation divided by mean) of several spatiotemporal parameters of gait obtained during the whole 6MWT (performed on a treadmill and investigated using a gait analysis laboratory); these ranged between 2% for stride time and 14% for step width, although the temporal patterns of such variations were not investigated. Interestingly, these authors showed, in a sub-analysis of participants with comparable walking speeds, that the mean values of spatiotemporal parameters of gait were similar between participants with COPD and healthy controls, but the variability of stride length, double support time, and step lengths seemed to be higher among cases. These results also extend our knowledge about the potential role of the variability of instrumental parameters as an early sign of deterioration of the gait pattern in persons with COPD, as already described for the general population of older adults [47,48,49].

### 4.3. Which Technology to Implement?

Several factors should be taken into consideration before implementing a technology for the instrumental evaluation of physical function in clinical or research practice. In the first place, size and cost may be considerably different from one instrument to another: tri-axial accelerometers are easily worn on different parts of the body and are generally affordable, whereas 3D gait analysis systems typically need dedicated spaces or infrastructures, and their cost is significantly higher. The physical test or function that needs to be objectively measured is another important factor for the choice of the system to implement: some gait labs are combined with treadmills that allow a detailed evaluation of gait or run, whereas force plates are often used both for balance tests’ evaluation (e.g., posturography) and assessment of strength or power during functional tests involving the lower limbs (e.g., jumps and chair stands). The capability to assess different functions with the same device may also be worth considering; among the application studies included in this systematic review, instrumented mattresses have been implemented only for the evaluation of gait, whereas force plates, accelerometers, NIRS, and sEMGs were employed to retrieve parameters from a variety of tests and function (i.e., gait, balance, jump, and replication of domestic activities). In addition, the aim of an instrumental evaluation of function should be clearly defined before the implementation of any technology. For example, in our study, we found that accelerometers have been used to obtain data about the intensity of gait (cadence), spatiotemporal parameters of gait (stride length, step length, and gait speed) and measures of gait symmetry. Such data cannot be retrieved using NIRS sensors or sEMGs which, in turn, have been implemented to obtain detailed information about the activation and usage of particular muscles (or muscular groups) during the performance of specific actions or functions.

### 4.4. Current Limitations to Technology Implementation

Our study highlights a significant implementation of technologies for the instrumental evaluation of physical function in persons with COPD over the last two decades. We also found that there was a significant heterogeneity in terms of the type of device used, applications, tested function(s), and retrieved metrics. This finding is of particular interest given that the lack of standardization and device-dependent assessment results were deemed as among the most important factors that hinder technology-aided assessments of physical function [50]. Furthermore, the possibility of deriving a general conclusion and giving precise indications regarding the application of these technologies to patients with COPD is limited by the fact that the studies included in our review recruited a limited number of participants. Indeed, 9 out of 24 papers included fewer than 20 participants with COPD. Furthermore, most authors selected their study population by excluding individuals affected by conditions potentially impacting the performance in physical function tests: cardiological, neurological, and musculoskeletal comorbidities were the most cited exclusion criteria. However, COPD is known to be frequently associated with multiple chronic diseases; in particular, due to shared risk factors, cardiological and neuro-vascular diseases are frequent comorbidities of COPD. These considerations suggest that the results from the studies included in this systematic review may not be directly generalizable to a significant share of older persons affected by this respiratory condition.

## 5. Conclusions

Novel technological solutions are more and more used to investigate physical function in persons affected by COPD, and a variety of potentially interesting metrics can be retrieved from such instrumental evaluation. However, a general lack of standardization and limitations in study design and sample size hinder the implementation of the instrumental evaluation of function in clinical practice.

## 6. Future Directions

Shared protocols for the validation of instrumental evaluation of function in persons affected by COPD are warranted. Future longitudinal studies should focus on the association between metrics derived from the instrumental evaluation of function and clinically meaningful outcomes for persons with COPD.

## Figures and Tables

**Table 1 sensors-22-06620-t001:** Characteristics of the included studies.

Study’s First Author	Year	Country	Study Design	Study Population	Subjects, *n* (% of Females)	Age, Mean (SD)	FEV_1_ % of Predicted, Mean (SD)
*Application studies*							
Annegarn [19]	2012	the Netherlands	Cross-sectional	Mixed: - COPD: outpatients recruited during a pre-rehabilitation assessment - Healthy subjects from previous trials conducted in the same centre	79 (40) 24 (37)	64 (9) 64 (6)	53.5 (18.7)
Beauchamp [11]	2012	Canada	Cross-sectional	Mixed: - COPD: outpatients - Healthy age–sex-matched controls	37 (54) 20 (60)	71 (7) 67 (9)	39.4 (16.3)
Canuto [12]	2010	Brazil	Cross-sectional	- COPD: outpatients	14 (NA)	69 (5)	39.4 (9.3)
Dos Reis [20]	2020	Brazil	Cross-sectional	Mixed: - COPD: outpatients - Healthy subjects	30 (33) 34 (32)	68 (8) 67 (8)	42.1 (16.4)
Fallahtafti [21]	2020	USA	Cross-sectional	Mixed: - COPD: outpatients - Healthy subjects from general population	17 (53) 23 (78)	64 (8) 60 (7)	NA
Gloeckl [22]	2017	Germany	Randomized Clinical Trial	COPD: Inpatients with COPD, GOLD stage III and IV	74 (32)	64 (9)	35.1 (10.1)
Iwakura [23]	2019	Japan	Cross-sectional	Mixed: - COPD: outpatients - Healthy subjects: age-matched, from local community centre	34 (0) 16 (0)	71 (8) 72 (6)	57.0 (28.0)
Janssens [24]	2014	Belgium	Cross-sectional	Mixed: - COPD: outpatients - Healthy subjects	18 (33) 18 (33)	65 (7) 64 (7)	51.0 (19.0)
Liu [25]	2019	the Netherlands	Cross-sectional	COPD: outpatients referred for pulmonary rehabilitation	44 (43)	62 (8)	55.9 (19.7)
Liu [10]	2020	USA	Cross-sectional	Mixed: - COPD: outpatients - healthy subjects	22 (41) 22 (73)	63 (9) 62 (9)	53.7 (18.5)
Liu [26]	2017	the Netherlands	Cross-sectional	Mixed: - COPD: outpatients referred for a pulmonary rehabilitation program in a specialized rehabilitation center - Healthy subjects from previous trials conducted in the same center	80 (40) 38 (37)	62 (7) 62 (6)	55.8 (19.4)
Marquis [27]	2009	Canada	Cross-sectional	Mixed: - COPD: outpatients - Healthy sedentary subjects	10 (10) 11 (9)	63 (6) 67 (6)	37.0 (13.0)
McCamley [28]	2017	USA	Cross-sectional	Mixed: - COPD: outpatients from the pulmonary clinical studies unit of university - Healthy elderly - Patients with bilateral peripheral artery disease	16 (NA) 25 (NA) 25 (NA)	64 (9) 66 (7) 64 (8)	NA
Meijer [29]	2014	the Netherlands	Cross-sectional	Mixed: - COPD: outpatients - healthy subjects	21 (24) 24 (29)	64 (8) 62 (6)	50.1 (20.1)
Morlino [30]	2017	Italy	Cross-sectional	Mixed: - COPD: outpatients - Healthy subjects	40 (28) 28 (43)	71 (7) 70 (7)	50.2 (21.1)
Munari [31]	2020	Brazil	Cross-sectional	- COPD: outpatients	36 (19)	67 (7)	51.1 (13.6)
Rutkowski [32]	2014	Poland	Cross-sectional	Mixed: - COPD: inpatients - Healthy individuals	33 (15) 48 (73)	66 (10) 59 (12)	NA
Terui [33]	2018	Japan	Cross-sectional	Mixed: - COPD: outpatients, who previously underwent pulmonary rehabilitation - Healthy individuals	16 (0) 26 (42)	71 (9) 68 (7)	58.4 (20.1)
Vaes [34]	2012	the Netherlands	Randomized crossover study	- COPD: outpatients, recruited during pre-rehabilitation assessment	21 (48)	64 (10)	42.0 (15.0)
Yentes [35]	2015	USA	Cross-sectional	Mixed - COPD outpatients recruited from local hospitals - Healthy subjects	17 (35) 21 (52)	64 (9) 65 (8)	50.2 (21.0)
Yentes [36]	2017	USA	Cross-sectional	Mixed - COPD individuals recruited from outpatients clinics - Healthy subjects	20 (20) 20 (55)	64 (10) 63 (8)	54.3 (19.2)
*Validation studies*							
Cheng [37]	2013	USA	Validation study	Mixed: - COPD: outpatients - Healthy subjects	6(83) 6(50)	NA	NA
Iwakura [23]	2019	Japan	Test-retest reliability	- COPD: outpatients	20 (0)	71 (8)	57.0 (28.0)
Liu [38]	2016	The Netherlands	Cross-sectional	Mixed: - COPD: outpatients (pre-rehabilitation assessment) - Healthy subjects	61 (38) 48 (53)	62 (7) 62 (6)	57.6 (20.0)
Sant’Anna [39]	2012	Brazil	Cross-sectional	- COPD: outpatients recently or currently enrolled in respiratory physiotherapy	30 (43)	67 (7)	44.0 (17.0)

COPD, chronic obstructive pulmonary disease; FEV1, forced expiratory volume in 1 s; GOLD, Global Initiative for Chronic Obstructive Lung Disease; NA, not available.

**Table 2 sensors-22-06620-t002:** Wearable devices employed and functions/parameters estimated in application studies.

Study’s First Author	Device	Protocol for Technology Application	Functional Test/Function	Parameter(s)	Values for COPD Participants, Mean (SD) *
Annegarn [19]	Accelerometer (Minimod, McRoberts, The Hague, The Netherlands), 100 Hz sampling frequency.	Accelerometer was attached to the trunk at the level of the sacrum.	6MWT	Walking intensity, counts·min^−1^ Cadence, strides·min^−1^ Anterior–Posterior AC, % Vertical AC, % Medio-Lateral AC, %	8658 (2971) 57 (6) 79.0 (10.7) 84.2 (10.2) 63.2 (14.0)
Canuto [12]	sEMG (analogical signals were amplified with 1000 gain. The signal was filtered with 10–500 Hz band-pass filter).	Electrodes positioned on the motor point of the rectus femoris, vastus lateralis, tibialis anterioris, and soleus during STS and 6MWT.	6MWT and STS	Muscle fatigue ACF during STS: Initial, degrees Final, degrees Muscle fatigue ACF during 6MWT: Initial, degrees Final, degrees	−11.6 (4.6) −18.3 (5.3) −11.9 (4.5) 14.5 (3.3)
Dos Reis [20]	sEMG (Myomonitor IV, DelSys, Boston, Massachusetts) at 2000 Hz. NIRS (OXYMON MK III, Artinis Medical System, Elst, The Netherlands) at 250 Hz.	Four muscle groups were assessed with EMG: sternocleidomastoid, Intercostal muscles, anterior deltoid, and trapezius. EMG signal was obtained for 6 min while the subject was performing the 6PBRT. NIRS was placed on intercostal muscles and anterior deltoid muscles.	6PBRT	Root mean square, mV: intercostal muscles sternocleidomastoid trapezius anterior deltoid Mean Frequency, Hz: intercostal muscles sternocleidomastoid trapezius anterior deltoid Oxyhemoglobin, Δ(O2Hb): intercostal muscles anterior deltoid deoxyhemoglobin, Δ(HHb): intercostal muscles anterior deltoid total hemoglobin, Δ(tHb): intercostal muscles anterior deltoid	Ranges 0.0046; 0.0051 0.0029; 0.0044 0.0543; 0.0587 0.073; 0.0844 54.85; 57.27 84.48; 88.08 73.17; 75.67 67.68; 73.03 −0.266; 0.357 −6.306; −2.58 −0.189; 0.169 6.757; 9.73 −0.494; 0.262 0.938; 7.051
Iwakura [23]	A tri-axial accelerometer system (Mimamori-gait system, LSI Medience Corporation, Japan)	The accelerometer was fixed to a belt around the level of the subject’s third lumbar vertebra.	Ten-meter walk test (14 m)	Gait speed, m·s^−1^ Step length, m Cadence, step·min^−1^ Walk ratio, mm·(steps·min^−1^)^−1^ Acceleration, g Step time SD, s	1.09 (0.22) 0.60 (0.08) 109 (10) 5.53 (0.69) 0.23 (0.08) 0.03 (0.01)
Marquis [27]	sEMG signals with a wireless amplifier system (TeleMyo2400T; Noraxon, Inc., Scottsdale, AZ), high pass filtered (10 Hz) and pre-amplified near electrodes. Band-pass filter 10–500 Hz and amplification at the receiver box.	sEMG signals from the soleus, tibialis anterior, medial gastrocnemius, vastus lateralis, and rectus femoris muscles of the right lower limb were measured during the 6 MWT.	6MWT (30-m long course according to the procedures recommended by ATS).	Median frequency, Hz: Soleus Tibialis anterior Gastrocnemius Vastus lateralis Rectus femoris Integrated EMG, µV: Soleus Tibialis anterior Gastrocnemius Vastus lateralis Rectus femoris	(Derived from figures) 85; 110 80; 90 85; 90 55; 70 50; 61 20,000; 25,000 30,000; 40,000 20,000; 25,000 12,000; 20,000 4000; 5000
Meijer [29]	Two tri-axial accelerometers (CIRO Activity Monitor (CAM); Maastricht Instruments B.V., Maastricht) and a Programmable Ambulant Signal AcQuisition system (PASAQ; Maastricht Instruments B.V.) for sEMG	A common ground electrode was placed on the ulnar styloid process. The cables from the electrodes were taped to the skin and placed into the PASAQ, which the participant wore in a small backpack.	Twelve domestic activities of daily life (cleaning windows, writing on a board, cleaning sink, pouring water and drinking, stretching arms, shaking hands, drawing picture, folding towels, putting towel on top shelf, walking, face care, and sweeping the floor).	Arm intensity, AU Arm elevation, AU Leg intensity, AU Relative muscle effort (trapezius), AU Relative muscle effort (biceps brachii), AU Relative muscle effort (deltoid), AU	Ranges 5.5; 70 −9.8; 19.1 1.6; 40.6 7.7; 52.1 3.1; 26.1 2.7; 35.7
Munari [31]	PortaMon NIRS device (Artinis Medical Systems).	NIRS was positioned on the vastus lateralis muscle of the dominant lower limb approximately 10 cm from the knee.	6-min step test (6MST): 20 -cm high step. Two trials performed with an interval of 30 min. Test was stopped once HR > 85% predicted max HR or SpO2 < 85% and resumed once the conditions for safe trial were met again.	Δ (difference between minute 6 –start): Oxyhemoglobin (O_2_Hb) Deoxyhemoglobin (HHb) Total hemoglobin (THb) Tissue saturation index (TSI), %	−5.40 (6.11) 7.73 (6.54) 2.33 (6.93) −7.34 (5.30)
Terui [33]	Wireless tri-axial accelerometer (MG-M1110; LSI Medience, Tokyo, Japan)	The accelerometer was fixed to a belt at the level of the subject’s L3.	10 m walk (1-m spare walkway area at the start and the end).	Difference in the absolute value for lateral acceleration. Difference between vertical acceleration when the right leg is in the stance phase and vertical acceleration when the left leg is in the stance phase. Lissajou index, %	0.22 (0.15) 0.15 (0.11) 34.2 (19.2)
Vaes [34]	Two tri-axial accelerometers (KXP94, Kionix Inc., Ithaca New York, USA) and the signal acquisition system for ambulant measurements (PASAQ, Maastricht Instruments B.V., Maastricht, The Netherlands)	Accelerometers were placed two fingers above the lateral malleolus of the right ankle and on the lower back and were connected with the PASAQ. Patients were randomly assigned to walk with rollator or modern draisine during the 6MWT.	6MWT (with rollator or modern draisine)	Strides, n: Modern draisine Rollator Stride length, m: Modern draisine Rollator Stride frequency, stides·s^−1^: Modern draisine Rollator RMS of the acceleration: Modern draisine Rollator	245.3 (60.9) 300.3 (49.1) 1.27 (0.14) 1.89 (0.73) 0.76 (0.14) 0.88 (0.11) 0.10 (0.03) 0.19 (0.07)

* Otherwise stated; Δ, delta change; 6MST, 6 min step test; 6MWT, 6 min walk test; 6PBRT, 6 min pegboard and ring test; AC, autocorrelation coefficient; ACF, angular coefficient of medium frequency; ATS, American Thoracic Society; NIRS, near-infrared spectroscopy; RMS, root mean square; sEMG, surface electromyography; STS, sit-to-stand test.

**Table 3 sensors-22-06620-t003:** Non-wearable devices employed and functions/parameters estimated in application studies.

Study’s First Author	Device	Protocol for Technology Application	Functional Test/Function	Parameter(s)	Values for COPD Participants, Mean (SD) *
Beauchamp [11]	Force plates (Advanced Medical Technology Inc.): two plates in parallel + one (in front of the subject. sEMG (gastrocnemius, tibialis anterior): pre-amplified signal at 500 gain + amplification by 1000. Signal digitally filtered from 20–250 Hz with 2nd-order dual pass Butterworth.	Force plates were used to capture footfall during perturbation-evoked reactions. sEMG was recorded bilaterally.	Perturbation-Evoked Reactions: subjects wore a harness with a cable attached posteriorly and were instructed to lean forward. Five perturbation trials were completed.	Foot-off time, ms Foot contact time, ms Swing time, ms APA duration, ms Integrated APA size, mm·ms	372 (78) 500 (89) 128 (28) 192 (52) 339 (253)
Fallahtafti [21]	Gait analysis (12-camera Raptor system, Motion Analysis Corp., Santa Rosa, CA, USA), using anteroposterior trajectory of retro-reflective marker attached to the right heel.	Retro-reflective spherical markers were attached bilaterally to lateral and medial metatarsophalangeal joint, base of the second toe, calcaneus, heel, lateral and medial malleoli, midshank, tibial tuberosity, lateral and medial knee joint centre, top of thigh, midthigh, greater trochanter, anterior and posterior superior iliac spine, and sacrum. Marker trajectories were analyzed for the last four minutes of each trial.	6MWT on a treadmill at self-selected walking speed (SSWS) + 1 slow and 1 fast (−20% and +20% SSWS) walking trials.	Step width, m: SSWS SSWS −20% SSWS +20% Step time, s: SSWS SSWS −20% SSWS +20% Step length, m: SSWS SSWS −20% SSWS +20%	0.09 (0.03) 0.09 (0.03) 0.02 (0.03) 0.84 (0.17) 0.71 (0.14) 0.69 (0.12) 0.42 (0.11) 0.45 (0.13) 0.52 (0.13)
Gloeckl [22]	Force platform (Leonardo Mechanograph^®^, Novotec Medical, Pforzheim, Germany) with 8 force sensors (800 Hz)	Postural balance and muscular power were assessed using the ground reaction force platform. The best test was used for analysis.	Postural balance (Romberg, semitandem, one foot beside and behind the other, and one-leg stance). Muscle power (two-legged jump).	Romberg APL (eyes closed), mm Semi-tandem APL (eyes closed), mm Semi-tandem APL (eyes open), mm One-leg stance APL (eyes open), mm Two-legged jump, W∙kg^−1^ Two-legged jump height, cm	429.50 (251.68) 885.50 (419.22) 365.50 (170.40) 839 (319.63) 24.30 (6.64) 23 (8.35)
Janssens [24]	Six-channel force plate (Bertec, OH, USA), sampled at 500 Hz, filtered using low-pass filter (5 Hz)	Participants sit barefoot on a stool on the force plate. The vision of the participants was occluded. Participants were asked to perform five STS movements.	5-STS	Sit duration, s Sit-to-stand duration, s Stand duration, s Stand-to-sit duration, s	0.87 (0.36) 0.14 (0.08) 1.79 (0.78) 1.08 (0.88)
Liu [25]	Three-dimensional motion analysis system with a dual-belt, instrumented treadmill and a virtual reality 180-degree projection screen (GRAIL, Motekforce Link, Amsterdam, the Netherlands) with integrated force plates (Forcelink, 12 channels, sample frequency 1000 Hz).	Patients performed a GRAIL-based 6MWTs on a split-belt, instrumented treadmill within a virtual reality environment.	6MWT, on treadmill	Mean stride time, s: Pre-PR Post- PR Mean stride length, m: Pre-PR Post- PR Mean step width, m: Pre-PR Post- PR Sample entropy stride length: Pre-PR Post- PR Sample entropy step width: Pre-PR Post- PR LDE CoMvel-ML: Pre-PR Post- PR LDE CoMvel-V: Pre-PR Post- PR LDE CoMvel-AP: Pre-PR Post- PR	1.02 (0.08) 1.00 (0.08) 1.45 (0.19) 1.48 (0.18) 0.18 (0.05) 0.18 (0.05) 1.17 (0.17) 1.21 (0.17) 1.43 (0.04) 1.43 (0.05) 2.83 (0.17) 2.77 (0.19) 2.78 (0.14) 2.79 (0.14) 2.75 (0.15) 2.70 (0.15)
Liu [10]	High-speed motion capture system (Motion Analysis, Santa Rosa, California) at 60 Hz	Retroreflective markers were placed on bony landmarks of the body, bilaterally. Participants were asked to walk on a treadmill at their SSWS. Three-dimensional marker data were used to calculate sagittal joint angle time series for the ankle, knee, and hip. The range of motion (RoM) was calculated for every right and left step from the joint angle time series	A total of 3.5 min at self-selected walking speed (SSWS), 1 trial at speeds 20% slow, and 1 trial at speed 20% fast—on treadmill.	Mean RoM, degrees: Ankle SSWS −20% SSWS SSWS +20% Knee SSWS −20% SSWS SSWS +20% Hip SSWS −20% SSWS SSWS +20% Sample entropy RoM: Ankle SSWS −20% SSWS SSWS +20% Knee SSWS −20% SSWS SSWS +20% Hip SSWS −20% SSWS SSWS +20% Local divergence exponent joint angle: Ankle SSWS −20% SSWS SSWS +20% Knee SSWS −20% SSWS SSWS +20% Hip SSWS −20% SSWS SSWS +20%	26.2 (5.9) 26.5 (6.0) 27.7 (6.0) 57.1 (8.8) 57.6 (8.7) 59.1 (7.2) 35.5 (5.6) 36.5 (5.1) 37.9 (4.9) 1.53 (0.38) 1.46 (0.40) 1.57 (0.51) 1.70 (0.42) 1.62 (0.39) 1.58 (0.36) 1.72 (0.23) 1.66 (0.23) 1.64 (0.29) 1.14 (0.11) 1.12 (0.17) 1.11 (0.15) 1.46 (0.14) 1.39 (0.16) 1.40 (0.17) 1.73 (0.18) 1.66 (0.18) 1.66 (0.19)
Liu [26]	Three-dimensional motion analysis system with a dual-belt, instrumented treadmill and a virtual reality 180-degree projection screen (GRAIL, Motekforce Link, Amsterdam, the Netherlands) with integrated force plates (Forcelink, 12 channels, sample frequency 1000 Hz).	Twenty five reflective markers were placed on anatomical landmarks of each participant. Each participant performed two 6MWT’s using the GRAIL.	6MWT, on treadmill	Cadence, steps·min^−1^ Double support time, s Stride time, s Stride length, m Step width, m	118.6 (10.3) 0.28 (0.04) 1.02 (0.09) 1.43 (0.18) 0.18 (0.04)
McCamley [28]	Three-dimensional marker trajectories (Motion Analysis Corp, Santa Rosa, CA; 60 Hz) and ground reaction forces (600 Hz; Kistler Group, Winterhur, Switzerland).	Thirty-three retro-reflective markers on specific anatomical locations.	Ten m walk: subjects walked over a 10 m path at their self-selected speed.	Peak angles, degrees Peak forces, N·kg^−1^ Peak moments, N·m·kg^−1^ Peak power, J·kg^−1^ Impulse, N·s·kg^−1^	Ranges 4.2 (4.6); 36.5 (6.8) 0.03 (0.02); 1.09 (0.09) −0.75 (0.28); 1.41 (0.15) −0.90 (0.35); 2.49 (0.50) −0.40 (0.16); 0.40 (0.16)
Morlino [30]	Instrumented mattress (GAITRite^®^, CIR Systems, USA)	Participants walked at comfortable speed along a 4 m-long instrumented mattress, four trials were evaluated.	4-m walk	Speed, cm·s^−1^ Cadence, step·min^−1^ Step length, cm Duration of the single-support, % Gait Cycle duration Duration of the double-support, % Gait Cycle duration	Derived from figures 100 110 57 38 25
Rutkowski [32]	Instrumented mattress (GAITRite^®^, CIR Systems, USA).	From the 5th meter, there was a four-meter GaitRite mat placed in the corridor. Analysis included 3 measurements taken at 3 points during the test duration.	6MWT: 30-m (evaluation on 4 m GaitRite)	Pace of gait, m·s^−1^ Stride length, cm Stride duration, s	156.6 (18.8) 74.8 (6.8) 0.48 (0.04)
Yentes [35]	High-speed motion capture system (Motion Analysis Corp., Santa Rosa, CA; 60 Hz) and piezoelectric force plate (Kistler Instrument Corp., Winterthur, Switzerland).	Reflective markers were placed on defined anatomical locations, bilaterally.	Ten m walk at normal pace. The subjects were asked to walk at normal pace (rest condition) or immediately after reporting breathlessness or muscle tiredness (provoked by treadmill walking with 10% incline) (no rest condition).	Speed, m·s^−1^: Rest No rest Step length, m: Rest No rest Step width, m: Rest No rest Step time, s: Rest No rest Stance time, s: Rest No rest Support time, s: Rest No rest Stride length, m: Rest No rest Stride time, s: Rest No rest	1.11 (0.17) 1.15 (0.18) 0.66 (0.06) 0.66 (0.06) 0.11 (0.04) 0.12 (0.04) 0.58 (0.06) 0.59 (0.06) 0.69 (0.09) 0.70 (0.10) 0.11 (0.03) 0.12 (0.04) 1.31 (0.13) 1.33 (0.13) 1.15 (0.11) 1.18 (0.13)
Yentes [36]	Infrared cameras (60 Hz; Motion Analysis Corp., Santa Rosa, CA)	A total of 3.5 min of walking on the treadmill at their self-selected pace and at two additional speeds (±20%).	Normal, fast, and slow walking (on treadmill)	Step length, m SSWS −20% SSWS SSWS +20% Step time, s SSWS −20% SSWS SSWS +20% Step width, m SSWS −20% SSWS SSWS +20%	0.449 (0.11) 0.479 (0.13) 0.543 (0.13) 0.790 (0.16) 0.670 (0.12) 0.646 (0.12) 0.094 (0.04) 0.097 (0.04) 0.096 (0.04)

* Otherwise stated; Δ, delta change; 6MWT, 6 min walk test; APA, anticipatory postural adjustment; APL, absolute path length; LDE, local divergence exponent; sEMG, surface electromyography; SSWS, self-selected walking speed; STS, sit-to-stand test.

**Table 4 sensors-22-06620-t004:** Parameters estimated from different devices.

Device	Parameters	Study’s First Author
Accelerometers	Cadence (steps/min)	Annegarn [19], Iwakura [23]
	Cadence (strides/min)	Annegarn [19], Vaes [34]
	Autocorrelation coefficient AP	Annegarn [19]
	Autocorrelation coefficient V	Annegarn [19]
	Autocorrelation coefficient ML	Annegarn [19]
	Gait speed	Iwakura [23]
	Step length	Iwakura [23]
	Step length/cadence (walk ratio)	Iwakura [23]
	Acceleration magnitude	Iwakura [23], Vaes [34]
	Step time	Iwakura [23]
	Intensity (upper limbs)	Meijer [29]
	Intensity (lower limbs)	Meijer [29]
	Relative muscle effort	Meijer [29]
	Difference in absolute ML acceleration	Terui [33]
	Difference between V acceleration in right stance and left stance	Terui [33]
	Lissajou index (symmetry evaluation)	Terui [33]
	Total amount of strides	Vaes [34]
	Stride length	Vaes [34]
Force plates	Foot-off time	Beauchamp [11]
	Foot contact time	Beauchamp [11]
	Swing time (foot-off time—foot contact time)	Beauchamp [11]
	APA	Beauchamp [11]
	Integrated APA size	Beauchamp [11]
	Absolute path length	Gloeckl [22]
	Peak W/kg during jump	Gloeckl [22]
	Jump height	Gloeckl [22]
	Sit duration in STS	Janssens [24]
	Sit-to-stand duration in STS	Janssens [24]
	Stand duration in STS	Janssens [24]
	Stand-to-sit duration in STS	Janssens [24]
Instrumented mattress	Speed	Morlino [30], Rutkowski [32]
	Step length	Morlino [30], Rutkowski [32]
	Cadence	Morlino [30]
	Single support duration	Morlino [30]
	Double support duration	Morlino [30]
	Stride duration	Morlino [30]
sEMG	Angular coefficient of medium frequency	Canuto [12]
	Mean frequency	Dos Reis [20]
	RMS frequency	Dos Reis [20]
	Median frequency	Marquis [27]
	Integrated frequency	Marquis [27]
NIRS	Δ (O_2_Hb)	Dos Reis [27], Munari [31]
	Δ (HHb)	Dos Reis [27], Munari [31]
	Δ (tHb)	Dos Reis [27], Munari [31]
Gait analysis/camera	Step width	Fallahtafti [21], Liu [25], Liu [26], Yentes [35], Yentes [36]
	Step duration	Fallahtafti [21]
	Step length	Fallahtafti [21], Yentes [35], Yentes [36]
	Stride time	Liu [25], Liu [26], Yentes [35]
	Stride length	Liu [25], Liu [26], Yentes [35], Yentes [36]
	Step time	Yentes [35]
	Stance time	Yentes [35]
	Stride sample entropy width	Liu [25]
	Stride sample entropy length	Liu [25]
	ROM	Liu [10]
	Sample entropy ROM	Liu [10]
	Local divergence exponent joint angle	Liu [10]
	Cadence (steps/min)	Liu [26]
	Double support time	Liu [26], Yentes [35]
	Speed	Yentes [35]
	Peak angles	McCamley [28]
	Peak forces	McCamley [28]
	Peak moments	McCamley [28]
	Peak power	McCamley [28]
	Impulse	McCamley [28]

Δ, delta change; AP, anterior–posterior direction; APA, anticipatory postural adjustment; ML, medio-lateral direction; V, vertical direction; NIRS, near-infrared spectroscopy; RMS, root mean square; ROM, range of motion; sEMG, surface electromyography; STS, sit-to-stand test.

**Table 5 sensors-22-06620-t005:** Devices employed, function(s) evaluated, parameters retrieved, and comparison metric(s) used in validation studies.

Study First Author	Year	Device	Test/Biomech fx	Parameter(s)	Comparison Device	Comparison Metric(s)	Comparison Value	Quality of the Study
Cheng [37]	2013	Phone app running on a Samsung Galaxy Ace	6MWT	Walking speed (estimated by SVM)	Clinical measurement	Root mean square error	Range 0.032; 0.133 (different SVM models)	Inadequate
Iwakura [23]	2019	A tri-axial accelerometer system (Mimamori-gait system, LSI Medience Corporation, Japan), 100 Hz sampling rate.	Tenm walk tst	Gait speed Step length Cadence Walk ratio Acceleration magnitude Step time	No	Intra-class correlation coef.: Gait speed (m·s^−1^) Step length (m) Cadence (step·min^−1^) Walk ratio Acceleration magnitude Step time SD	ICCs (95%CI) 0.97 (0.93–0.99) 0.97 (0.92–0.99) 0.96 (0.90–0.98) 0.97 (0.92–0.99) 0.97 (0.92–0.99) 0.91 (0.79–0.96)	Doubtful
Liu [38]	2016	Three-dimensional motion analysis system with a dual-belt, instrumented treadmill and a virtual reality 180-degree projection screen (GRAIL, Motekforce Link, Amsterdam, the Netherlands) with integrated force plates (Forcelink, 12 channels, sample frequency 1000 Hz).	6MWT	Walking speed	Clinical evaluation (overground 6MWT)	Intra-class correlation coefficient	ICCs (95%CI) 0.74 (0.51–0.86)	Doubtful
Sant’Anna [39]	2012	Power Walker 610 (Yamax, 1-5-7, Chuo-cho, Meguro-ku, Tokyo 152-8691 Japan): pedometer combined with accelerometer.	Walking protocol	Number of steps (n) Walking distance (m) Intensity (m/min) Energy expenditure (Kcal)	Video recording and SenseWear Armband (for energy expenditure estimation)	Pearson correlation coefficient: Number of steps -fast -slow Walking distance -fast -slow Walking intensity (speed) -fast -slow Energy expenditure -fast -slow	rho 0.95 0.79 0.48 0.63 0.47 0.61 0.83 0.65	Doubtful

6MWT, 6 min walk test; CI, confidence interval; COPD, chronic obstructive pulmonary disease; ICC, intraclass correlation coefficient SD, standard deviation; SMV, support vector machine.

## Data Availability

No new data were created or analyzed in this study. Data sharing is not applicable to this article.

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
