# Peer review of "Technologies for the Instrumental Evaluation of Physical Function in Persons Affected by Chronic Obstructive Pulmonary Disease: A Systematic Review"

_sensors, 2022, doi:10.3390/s22176620_

Round 1

Reviewer 1 Report

This manuscript is well written to describe current trends to assess physical function in patients with COPD using objective systems. However, it is still not clear why the assessment of physical function is valuable, and which parameters are primary in this population. For instance, most of the previous studies conducted gait analysis for patients with COPD, but I am not sure which factors contribute to their worse gait functions. Please introduce essential characteristics of physical function in the patients with COPD. 

Minor comment

Table 2 is too huge and poor readability.

I highly recommend to divide several tables according to the devices or tests.

Author Response

We are very grateful for the reviews provided by the editor and each of the external reviewers of this manuscript. The comments are encouraging, and the reviewers’ suggestions helped us to improve the quality of the study. Corrections made in the text have been highlighted in blue. Please see our explanations to the comments as follow:

Comments from Reviewer 1

COMMENT 1: This manuscript is well written to describe current trends to assess physical function in patients with COPD using objective systems. However, it is still not clear why the assessment of physical function is valuable, and which parameters are primary in this population. For instance, most of the previous studies conducted gait analysis for patients with COPD, but I am not sure which factors contribute to their worse gait functions. Please introduce essential characteristics of physical function in the patients with COPD.

RESPONSE: Thank you for this insightful comment: we have added the following sentence in the introduction on page 1 and 2 (lines 43 to 48): “Several mechanisms may impact on the physical performance of persons with COPD: for example, impaired ventilatory mechanics (such as dynamic hyperinflation), modification of the ventilation-perfusion relationship and hypoxemia, pulmonary hyper-tension and other cardiovascular factors. The 6-minute walking test (6MWT) is currently implemented to measure the impact of the COPD on physical performance.”. In addition, we have modified the sentence on page 2, lines 48 to 51 as follows: “However, physical function tests may also help to…of health”.

COMMENT 2: Table 2 is too huge and poor readability. I highly recommend to divide several tables according to the devices or tests.

RESPONSE: Thank you for this suggestion. As suggested, we have divided Table 2 in two tables (Table 2 and Table 3) according to the portable characteristic of the device. In addition, we have renamed the other tables in the main document.

Sincerely,

The Authors

Reviewer 2 Report

Dear corresponding Author I appreciated your paper

Author Response

We are very grateful for the reviews provided by the editor and each of the external reviewers of this manuscript. The comments are encouraging, and the reviewers’ suggestions helped us to improve the quality of the study. Corrections made in the text have been highlighted in blue. Please see our explanations to the comments as follow:

Comments from Reviewer 2

COMMENT 1: Dear corresponding Author I appreciated your paper.

RESPONSE: Thank you for your favourable comment.

Sincerely,

The Authors

Reviewer 3 Report

This is a well-structured systematic review, so I do not have any problems with the approach.

This is a systematic review paper.

It does not have a research question as such, but more looks at a specific topic and see what has been done and the topic is relevant for sensors Journal.

It does not add new knowledge but somewhat summarizes the existing knowledge in a systematic way.

The methodology is standard and the authors do not develop any new methodology and the conclusions are based on previous work.

Author Response

We are very grateful for the reviews provided by the editor and each of the external reviewers of this manuscript. The comments are encouraging, and the reviewers’ suggestions helped us to improve the quality of the study. Corrections made in the text have been highlighted in blue. Please see our explanations to the comments as follow:

Comments from Reviewer 3

COMMENT 1: This is a well-structured systematic review, so I do not have any problems with the approach.

This is a systematic review paper.

It does not have a research question as such, but more looks at a specific topic and see what has been done and the topic is relevant for sensors Journal.

It does not add new knowledge but somewhat summarizes the existing knowledge in a systematic way.

The methodology is standard and the authors do not develop any new methodology and the conclusions are based on previous work.

RESPONSE: Thank you for your favourable comments. As you suggested, this systematic review was not conceived to add new knowledge to the field of respiratory medicine, but rather to provide, summarized in a systematic way, an overview of the implementation of new technologies for the assessment of physical function in persons with COPD. Potentially, this study could also allow future readers to retrieve reference values from prior studies about the instrumental evaluation of physical function in this population.

Sincerely,

The authors